# The Deep-MCL Line: A Reliable Anatomical Landmark to Optimize the Tibial Cut in UKA

**DOI:** 10.3390/jpm13050855

**Published:** 2023-05-19

**Authors:** Sébastien Parratte, Jeremy Daxhelet, Jean-Noel Argenson, Cécile Batailler

**Affiliations:** 1Department of Orthopaedic Surgery, International Knee and Joint Centre, Hazza Bin Zayed St., Abu Dhabi P.O. Box 46705, United Arab Emirates; 2Department of Orthopedics and Traumatology, St. Marguerite Hospital, Aix Marseille University, Institute of Movement and Locomotion, 270 Bd de Sainte-Marguerite, 13009 Marseille, France; jean-noel.argenson@ap-hm.fr; 3Department of Orthopaedic Surgery, Clinique Saint-Luc Bouge, Rue Saint-Luc 8, 5004 Namur, Belgium; daxheletjeremy@gmail.com; 4Department of Orthopaedics, Croix Rousse Hospital, University of Lyon 1, 69004 Lyon, France; 5Claude Bernard Lyon 1 University, LBMC UMR_T9406, 69100 Lyon, France

**Keywords:** medial unicompartmental knee arthroplasty, anatomical landmarks, coronal alignment, tibial slope, deep medial collateral ligament

## Abstract

The extramedullary guides for the tibial resection during medial unicompartmental knee arthroplasty (UKA) are inaccurate, with an error risk in coronal and sagittal planes and cut thickness. It was our hypothesis that the use of anatomical landmarks for the tibial cut can help the surgeon to improve accuracy. The technique described in this paper is based on the use of a simple and reproducible anatomical landmark. This landmark is the line of insertion of the fibers of the deep medial collateral ligament (MCL) around the anterior half of the medial tibial plateau called the “Deep MCL insertion line”. The used anatomical landmark determines the orientation (in the coronal and sagittal planes) and the thickness of the tibial cut. This landmark corresponds to the line of insertion of the fibers of the deep MCL around the anterior half of the medial tibial plateau. A consecutive series of patients who underwent primary medial UKA between 2019 and 2021 were retrospectively reviewed. A total of 50 UKA were included. The mean age at the time of surgery was 54.5 ± 6.6 years (44–79). The radiographic measurements showed very good to excellent intra-observer and inter-observer agreements. The limb and implant alignments and the tibial positioning were satisfying, with a low rate of outliers and good restoration of the native anatomy. The landmark of the insertion of deep MCL constitutes a reliable and reproducible reference for the tibial cut axis and thickness during medial UKA, independent of the wear severity.

## 1. Introduction

Unicompartmental knee arthroplasty (UKA) remains a demanding surgical procedure, and optimal implant positioning is essential to obtain satisfactory outcomes [1,2,3]. Various complications can occur after UKA, such as implant malpositioning, malalignment, and implant over- or under-sizing. Sub-optimal implant position can cause the failure of the UKA, with potential complications such as persistent pains or tibial component loosening [1,2,3].

As several systems use the tibial resection as a reference for the femoral resection, optimizing the tibial cut is a crucial step of the surgery. This is even more important as the risk of implant malpositioning concerns mainly the tibial implant, with a risk of outliers in the coronal and sagittal alignment and a risk of excessive tibial resection [4,5]. The extramedullary guides for the tibial resection have been improved over time, but several studies reported a persistent high percentage of outliers [5,6,7] with a risk of error in coronal and sagittal planes. Assistive technologies, such as robotic-assisted systems, have been developed to improve the accuracy of bone resections and implant positioning in UKA [8,9,10] with promising results [6,11,12,13,14]. Due to the high cost of these devices, however, less than 1% of the surgeons in the world have access to a robotic-assisted system for UKA. Therefore, it was our hypothesis that the use of simple anatomical landmarks for bone tibial bone resection can be reliable and help surgeons to improve tibial cut accuracy with conventional instrumentation. Several studies have described and assessed bony landmarks for the tibial rotation in UKA [15,16,17]; however, to our knowledge, no study has described the use of tibial anatomical landmarks for the orientation and the thickness of the tibial cut in medial UKA.

Therefore, the aims of this paper were as follows: (1) to describe the surgical technique of the tibial cut in UKA using the tibial insertion of the fibers of the deep medial collateral ligament (MCL) as a landmark for frontal and sagittal orientation and the thickness of resection; (2) to assess the accuracy of the tibial cut with this surgical technique as measured on post-operative radiographs (MPTA, tibial slope, joint line height, and HKA angle).

## 2. Materials and Methods

### 2.1. Surgical Technique

The medial UKA is indicated for osteoarthritis without severe constitutional deformity. The UKA principles are to compensate for the wear, respect the anatomy of the proximal tibial epiphysis, and perform a pure resurfacing surgery, respecting the ligamentous envelope. Based on the literature, angular limits of resection can be comprised between 0 and 5 degrees of varus for the frontal plane and between 2 and 6 degrees of posterior slope for the sagittal plane. Following the standard surgical technique, the tibial cut is performed using a conventional extramedullary guide, the surgeon aiming for the ideal position of the jig to reach the goals of resection in terms of thickness of resection and frontal/sagittal orientation based on its own judgment (Figure 1). Fine adjustments of the cut axis can be challenging, and this might explain the degrees of inaccuracy observed with conventional instrumentation in the literature.

The technique described in this paper is based on the use of a simple and reproducible anatomical landmark. This landmark is the line of insertion of the fibers of the deep MCL around the anterior half of the medial tibial plateau called the “Deep MCL insertion line”. The visualization of this line is relatively simple when performing a very conservative approach exposure (without any release of the medial tibial plateau) and after the removal of the anterior osteophytes. For this technique, two points are marked along the insertion of the deep MCL, and then the line joining these two points is drawn on the bone using the electrocautery knife. The used anatomical landmark determines the orientation (in the coronal and sagittal planes) and the thickness of the tibial cut. The cutting jig can then be directly aligned onto this line and pinned, the lower part of the jig being used only as a support of the cutting jig. A second check can be performed once the cutting jig is set up.

A medial subvastus approach without any medial or lateral release is performed. The deep MCL insertion is visualized, and two points (one anterior and one more posterior) are marked along its insertion around the anterior part of the medial tibial plateau (Figure 2a). These two points are used to draw a line which is usually just below the medial osteophytes. These medial osteophytes can be partially removed to better visualize the insertion of the fibers around the tibial plateau if needed (Figure 2b). To remove these osteophytes while avoiding any damage to the MCL insertion, a small Hohmann retractor can be used. Following the line of the MCL insertion around the medial tibial plateau and extending this line anteriorly can accurately guide the frontal and sagittal orientation of the cut (posterior slope) and the thickness of resection. Indeed, the medial osteophyte is frequently used as a landmark medially but shows only the thickness of the cut medially and anteriorly. It is thus insufficient to avoid a valgus cut compared to the line that has been described earlier (Figure 3). As the level of the insertion of the deep MCL is fixed and not related to the severity of the wear or osteoarthritis, this landmark can reliably be used to determine the thickness of the cut. The use of a tibial stylus, whose size varies significantly with the severity of the wear, is thus not necessary. The tibial rotation is determined as usual, drawing a line between the point considered the medial point to the tibial insertion of the anterior cruciate ligament and the most anterior point of the medial tibial plateau. This line is parallel to the lateral facet of the medial condyle.

When the cutting guide can be positioned on the desired cut axis and set in place (Figure 4). The tibial cut is performed with the saw as usual. To confirm the cut axis and thickness, the tibial resection should have almost no attachment with the articular capsule (cut inside the ligamentous envelope) (Figure 5). The final implant is positioned in the ligamentous envelope, preserving the deep MCL insertion and the joint line height. The meniscal scar was used to evaluate the restoration of the joint line, the upper level of the polyethylene insert being exactly at the level of the meniscal scar (Figure 6).

### 2.2. Patients

After obtaining ethics internal review board approval, a consecutive series of patients who underwent a primary medial UKA between 2019 and 2021 at a single institution were retrospectively reviewed. The indication for surgery was medial femorotibial osteoarthritis or femoral osteonecrosis, with a reducible deformation and without anterior laxity. Exclusion criteria were incomplete data (radiographs) and previous tibial osteotomy. Of the 59 primary UKA performed during this period, 50 met the criteria (6 lateral UKA and 3 patients with incomplete radiographs). The mean age at the time of surgery was 54.5 ± 6.6 years (44–79). Mean BMI was 32.7 ± 3.7 kg/m^2^ (27–44). A total of 40% (*n* = 20) were male patients, and 48% (*n* = 24) were operated on the left knee. A total of 24% had a grade 4, and 76% had a grade 3 of medial femorotibial osteoarthritis (Kellgren Lawrence). All UKA were performed using conventional instrumentation by a single senior surgeon with 15 years of experience in UKA. All patients received the same cemented morphometric fixed-bearing medial UKA (Persona Partial Knee System, Zimmer Biomet, Warsaw, IN, USA) [18]. This system is a tibia-based technique using the spacer-block technique for femoral preparation [18,19]. The tibial cut is thus an essential factor in the quality of the distal femoral cut and the entire procedure.

### 2.3. Data Assessment

The radiographic assessment was performed preoperatively and at 2 months, including an anteroposterior view, lateral view of the knee, and a long-leg standing radiograph performed according to a standardized protocol in the same radiological center. Standardized radiographic measurements were performed: HKA angle, mechanical Medial Distal Femoral Angle (mMDFA), Medial Proximal Tibial Angle (MPTA), tibial slope, the joint line height, the Cartier angle, the coronal axis, and the thickness of the tibial cut. Restitution of joint line height was assessed using the two methods of Weber [20]. The tibial resection was measured with the technique described by Negrin [21]. The radiographs were calibrated, allowing an accurate measurement up to 0.1 mm. Radiological measurements were performed twice by two independent reviewers (CB and JD) for all measurements to assess the reliability of each measurement. The thickness of the polyethylene insert was reported in the surgical report.

### 2.4. Statistical Analysis

Statistical analysis was performed using the XL STAT software (Version 2021.2.1, Addinsoft Inc., Paris, France). Data were described using means, standard deviation, ranges for continuous variables, and counts (percent) for categorical variables. The intra- and inter-observer reliabilities of the radiographic measurements were evaluated by an intraclass correlation coefficient. Strength of agreement for the kappa coefficient was interpreted as follows: <0.20 = unacceptable, 0.20–0.39 = questionable, 0.40–0.59 = good, 0.60–0.79 = very good, and 0.80–1 = excellent [22].

## 3. Results

The radiographic measurements showed very good to excellent intra-observer and inter-observer agreements (Table 1). The limb and implant alignments and the tibial positioning are reported in Table 2. The tibial insert was 8 mm for 50% of the patients (*n* = 25), 9 mm for 48% (*n* = 24), and 10 mm for 2% (*n* = 1).

## 4. Discussion

The tibial cut is a challenging step during medial UKA. This surgical technique based on a bony landmark aims to reduce the error risk of the tibial cut axis. This technique was performed for many years by the senior surgeon with satisfying results and appears safe and reliable.

Several limitations should be outlined in our study. This study was not comparative with other surgical techniques (robotic-assisted or manual with extramedullary guide). There were no functional outcomes or long-term data. Nevertheless, this study aimed to describe for the first time the surgical technique of the tibial cut in UKA using the tibial insertion of the fibers of the deep MCL and its accuracy. A long-term comparative study would be interesting to perform secondarily.

Tibial malpositioning is one of the most common errors during medial UKA [6]. The risk is to perform the tibial cut in the valgus or varus compared to the epiphyseal axis. The mean axis of the tibial cut was satisfying in this study (87.7° ± 1.6°), with only one patient with a tibial cut axis superior to 5° of varus (84°). The mean difference between the tibial cut axis and the tibial epiphyseal axis was inferior to 0.6° ± 1.1. Two main philosophies for positioning UKA components are described in the literature [23]. The mechanical alignment technique references the mechanical axis of long bones and makes frontal bone cuts perpendicular to them [24]. This technique is easier to perform than the conventional technique because the tibial cut is performed at 90° of the tibial mechanical axes. An alternative alignment technique was popularized by Cartier who tried to reproduce the tibial epiphyseal axis with the tibial cut in the coronal plane [25,26]. A threshold value is recommended with a tibial cut inferior to 5° of varus compared to the tibial mechanical axis. This last philosophy avoids a valgus cut compared to the epiphyseal axis with a risk of loosening or secondary subsidence due to the soft bone in the lateral part of the tibial resection [23]. To reproduce the tibial epiphyseal anatomy also aims to obtain a perfect congruence between the femoral implant and the plateau surface and avoid a position on the edges of the condylar implant. However, to perform a tibial cut with some degree of varus with an extramedullary guide is difficult and inaccurate. The rate of outliers in the coronal plane after conventional medial UKA is significant in the literature up to 35% [4,6]. To limit the number of outliers in UKA, robotic surgery has been developed, but its access remains limited to only a subset of surgeons, and its cost-efficiency is still to be demonstrated. The landmark described in this study is a simple, cost-efficient additional control to improve the accuracy of the tibial resection and reduce the number of outliers. The advantage of this technique compared to the medial osteophyte landmark is the line following the insertion of deep MCL determines the coronal and sagittal planes (Figure 3). The quality of the tibial cut also determines the femoral implant positioning. Most of the UKA surgical techniques have femoral cuts dependent on the tibial cut. If the alignment of the tibial cut is not satisfying, the femoral implant has a risk of malpositioning.

In this study, the joint line height was distalized at a mean of 0.9 mm ± 1.1 compared to the pre-operative X-rays, probably due to the pre-operative wear of the femoral condyle. Taking this point into consideration, the restitution of the joint line height using this tibial landmark was thus satisfying with the smallest insert sizes (8 or 9 mm). Restitution of joint line height after UKA, and particularly avoidance of excessive tibial resection, has a major impact on patients’ outcomes and tibial implant survivorship [27,28,29]. In addition to making the tibial implant rest on more fragile cancellous bone, excessive tibial cutting also leads to shifting the contact point of the femoral component towards the periphery of the tibial plateau due to the plateau’s funnel shape. A biomechanical study demonstrated that after UKA, the mean strain on the proximal tibial cortex increased by 6%, 13%, and 18% when tibial resection levels of 2 mm, 4 mm, and 6 mm were modeled, respectively [29]. Another study demonstrated similar results: 4 mm increased distal resection increased tibial strain variance by 35% [27]. An excessive tibial cut can also lead to a distalization of the femoral implant by dependent cuts, lower the joint line height in the medial compartment, and result in a no-anatomical oblique joint line [26]. The improvement of the joint line height can reduce the polyethylene wear, the loosening risk, and the progression of osteoarthritis in the contralateral compartment [1,30]. A reduction in tibial resection may also improve some tibial pain due to the excessive strain on the proximal tibial cortex [27]. The landmark of the insertion of deep MCL constitutes a stable reference for the cut thickness, independent of the wear severity and the position of the tibial sizer. This landmark can increase the reproducibility of the tibial resection, and this might be particularly helpful and cost-efficient for surgeons with a low volume of UKA.

## 5. Conclusions

The results of our study confirmed that the deep-MCL line is a reliable anatomical landmark to optimize the tibial cut in UKA. This landmark corresponds to the line of insertion of the fibers of the deep MCL around the anterior half of the medial tibial plateau. The deep-MCL line can help surgeons to improve the accuracy and the reproducibility of the tibial cut in UKA for both the coronal and the sagittal plans. This technique can help to reduce the outliers without the extra cost related to the use of assistive computer-assisted technologies. 

## Figures and Tables

**Figure 1 jpm-13-00855-f001:**
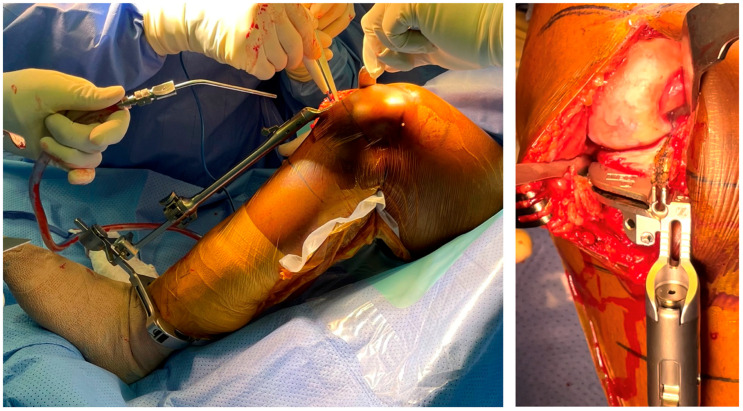
Extramedullary guide to check the tibial slope and the coronal axis for the tibial cut.

**Figure 2 jpm-13-00855-f002:**
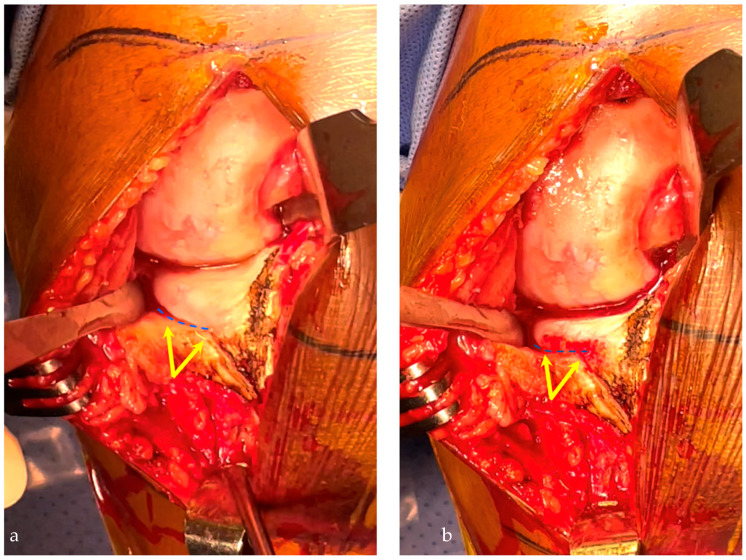
Insertion of the deep medial collateral ligament (MCL) on the medial proximal tibial plateau (yellow arrows), which delineates the tibial cut axis (blue line) (**a**). The osteophytes resection improves the visualization of this landmark (**b**).

**Figure 3 jpm-13-00855-f003:**
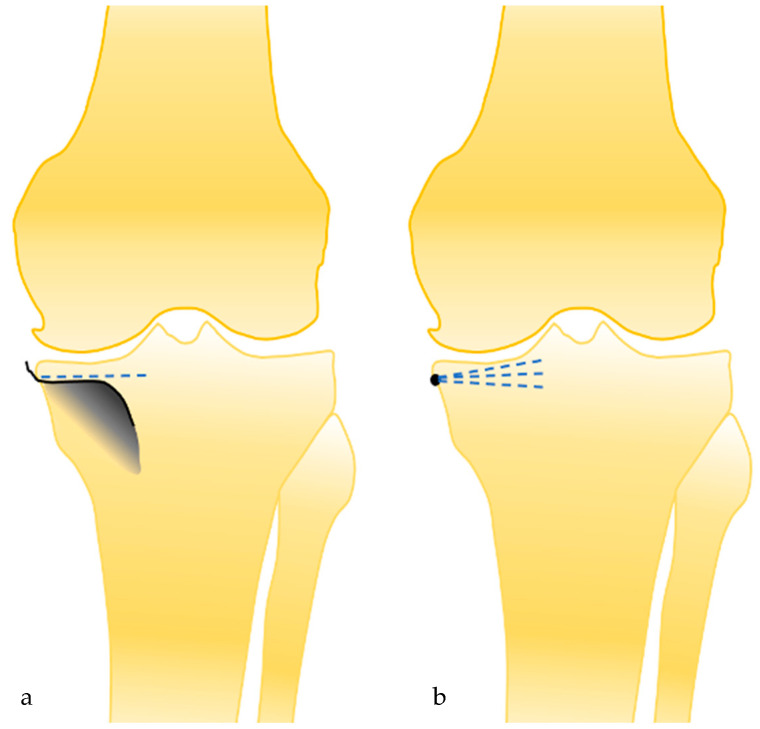
The deep MCL landmark allows us to determine the coronal axis of the tibial cut (**a**), and not only the height of the tibial cut medially, as the medial osteophyte landmark (**b**).

**Figure 4 jpm-13-00855-f004:**
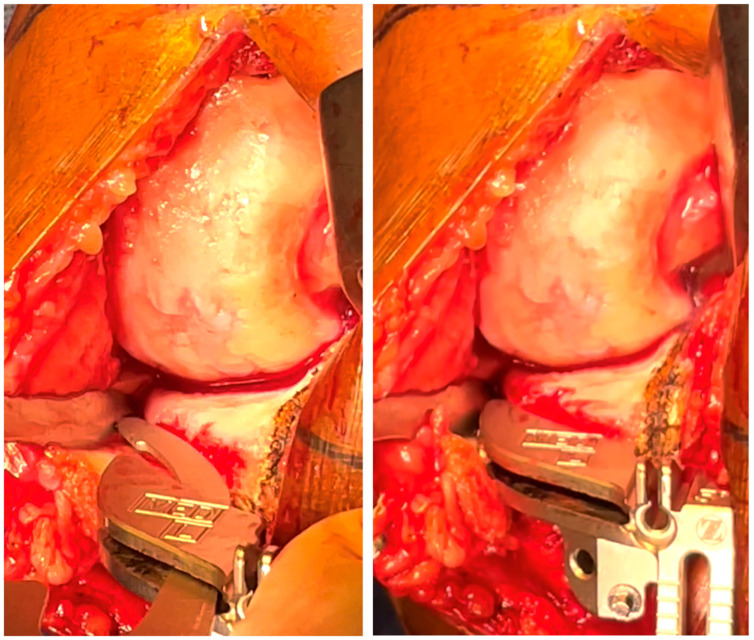
After the visualization of the tibial landmark, the cutting guide is positioned directly on this landmark and then set to the bone.

**Figure 5 jpm-13-00855-f005:**
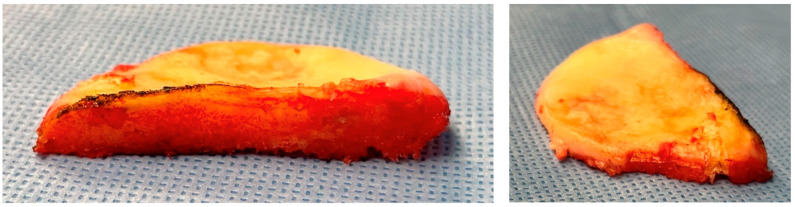
The tibial resection allows to confirm if the cut axis is satisfying: the cut tightness should be similar anteriorly and posteriorly, and the cut should be at the limit of the capsular attaches.

**Figure 6 jpm-13-00855-f006:**
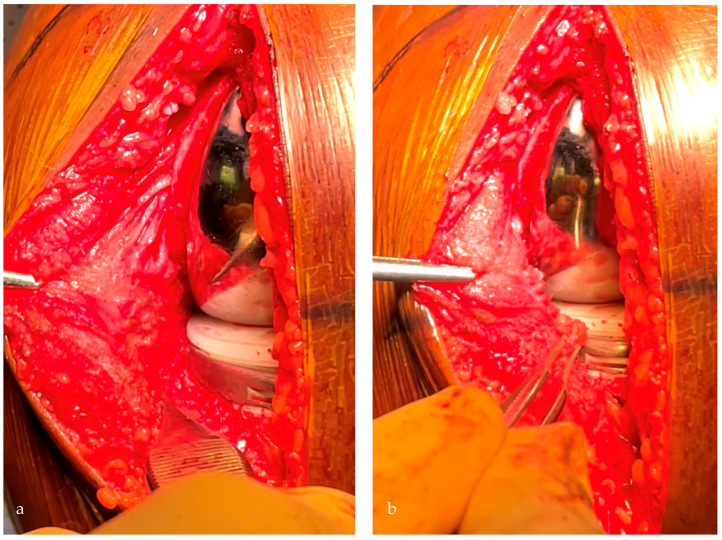
With the definitive implants, the tibial implant should be positioned in the soft tissues’ envelope respecting the deep MCL (**a**) and restoring the joint line height, visualized with the meniscal scar level (**b**).

**Table 1 jpm-13-00855-t001:** Intraobserver and interobserver coefficients for the radiographic measurement.

	Intra Observer ICC	Inter Observer ICC	Agreement
HKA angle	0.98	0.98	Excellent
mMDFA	0.95	0.92	Excellent
MPTA	0.90	0.83	Excellent
Tibial slope	0.82	0.83	Excellent
Cartier angle	0.85	0.69	Very good
Joint line height	0.85	0.72	Very good
Tibial cut height	0.80	0.75	Very good
Tibial cut Coronal Axis	0.87	0.78	Very good

Strength of agreement for the kappa coefficient was interpreted as follows: <0.20 = unacceptable, 0.20–0.39 = questionable, 0.40–0.59 = good, 0.60–0.79 = very good, and 0.80–1 = excellent.

**Table 2 jpm-13-00855-t002:** Preoperative and postoperative radiographic measurements and outliers.

	Preoperative Data N = 50	Postoperative Data N = 50
HKA (°)	173.5 ± 3.6	176.5 ± 3.1
(mean ± SD) [Min; Max]	[164.6; 180]	[170; 185]
mMDFA (°)	91.2 ± 2.2	92.2 ± 2.3
(mean ± SD) [Min; Max]	[87; 96]	[88; 96]
MPTA (°)	86.4 ± 1.5	86.8 ± 1.5
(mean ± SD) [Min; Max]	[83; 89]	[84; 90]
OUTLIERS MPTA < 85°	6 (12%)	1 (2%)
Slope (°)	80.9 ± 3.2	82.6 ± 2.3
(mean ± SD) [Min; Max]	[74; 87]	[78; 87]
OUTLIERS Slope < 78°	7 (14%)	0
Cartier angle (°)	2.6 ± 2.8	-
(mean ± SD) [Min; Max]	[−3; 7]	-
Joint line height (femoral cortex) (mm)	-	0.9 ± 1.1
(mean ± SD) [Min; Max]		[−1.7; 4.5]
Joint line height (femoral diaphysis) (mm)	-	0.8 ± 1.1
(mean ± SD) [Min; Max]		[−1.7; 4.5]
OUTLIERS Joint line height > 2 mm	-	3 (6%)
Tibial resection height (mm)	-	6.0 ± 1.7
(mean ± SD) [Min; Max]		[1; 9.5]
Tibial cut axis (°)	-	87.7 ± 1.6
(mean ± SD) [Min; Max]		[84; 92]
OUTLIERS Tibial cut axis < 85°	-	1 (2%)
OUTLIERS Tibial cut axis > 90°	-	2 (4%)
Difference between tibial cut and Cartier angle (°)	-	0.57 ± 1.1
(mean in absolute value ± SD) [Min; Max]		[−5; 4]

HKA: Hip Knee Ankle angle; mMDFA: mechanical Medial Distal Femoral Angle; MPTA: Medial Proximal Tibial Angle; JLCA: Joint Line Convergence Angle; JLO: Joint Line Orientation; SD: Standard Deviation.

## Data Availability

Data is unavailable due to ethical restrictions.

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
