# Peer review of "The Deep-MCL Line: A Reliable Anatomical Landmark to Optimize the Tibial Cut in UKA"

_jpm, 2023, doi:10.3390/jpm13050855_

Round 1
Reviewer 1 Report
This was an excellent paper with very good photographs to demonstrate their tip and technique when performing the tibial cut for medial unicondylar arthroplasty. The only critique I have is that there is no control group and it would have been nice to compare this to either robotic assisted UKA or manual UKA with the stylus. This might be something to consider for a future study - compare 1 to 1 with a group of manual UKA using the stylus technique or robotically assisted UKA.
Author Response
Thank you for your advice. We will perform this comparative study in the future.
Reviewer 2 Report
It is an interesting and well-presented study. there are a few suggestions that the authors can do in the revision
1) add the limitations of the study in the discussion section.
2) More details on the pain scores and improvement in quality of life can be studied in the subsequent long term study.
Author Response
1) Thank you for your comment. We have added the limitations in the discussion.
2) In a second study, we will assess the long-term functional outcomes.